# Host-pathogen coevolution increases genetic variation in susceptibility to infection

Elizabeth ML Duxbury[1,2†], Jonathan P Day[1†], Davide Maria Vespasiani[1], Yannik Thüringer[1], Ignacio Tolosana[1], Sophia CL Smith[1], Lucia Tagliaferri[1], Altug Kamacioglu[1], Imogen Lindsley[1], Luca Love[1], Robert L Unckless[3], Francis M Jiggins[1]*, Ben Longdon[1,4]*

[1]Department of Genetics, University of Cambridge, Cambridge, United Kingdom; [2]School of Biological Sciences, University of East Anglia, Norwich, United Kingdom; [3]Department of Molecular Biosciences, University of Kansas, Lawrence, United States; [4]Centre for Ecology and Conservation, Biosciences, University of Exeter (Penryn Campus), Cornwall, United Kingdom

*For correspondence:
fmj1001@cam.ac.uk (FMJ);
b.longdon2@exeter.ac.uk (BL)

†These authors contributed equally to this work

**Competing interests:** The authors declare that no competing interests exist.

**Abstract** It is common to find considerable genetic variation in susceptibility to infection in natural populations. We have investigated whether natural selection increases this variation by testing whether host populations show more genetic variation in susceptibility to pathogens that they naturally encounter than novel pathogens. In a large cross-infection experiment involving four species of *Drosophila* and four host-specific viruses, we always found greater genetic variation in susceptibility to viruses that had coevolved with their host. We went on to examine the genetic architecture of resistance in one host species, finding that there are more major-effect genetic variants in coevolved host-pathogen interactions. We conclude that selection by pathogens has increased genetic variation in host susceptibility, and much of this effect is caused by the occurrence of major-effect resistance polymorphisms within populations.
DOI: https://doi.org/10.7554/eLife.46440.001

## Introduction

From bacteria to plants and insects to humans, it is common to find considerable genetic variation in susceptibility to infection in natural populations (*Chapman and Hill, 2012*; *Bangham et al., 2008a*; *Hammond-Kosack and Jones, 1997*; *Lazzaro et al., 2004*). This variation in susceptibility can determine the impact of disease on health and economic output (*Cooke and Hill, 2001*; *King and Lively, 2012*; *Alonso-Blanco and Méndez-Vigo, 2014*; *Burgner et al., 2006*). In nature and breeding programs, it determines the ability of populations to evolve resistance to infection. Insect populations, like those of other organisms, typically contain considerable genetic variation in susceptibility to infection (*Bangham et al., 2008a*; *Lazzaro et al., 2004*; *Tinsley et al., 2006*; *Obbard and Dudas, 2014*), and provide a convenient laboratory model in which to investigate basic questions about how this variation is maintained (*Magwire et al., 2012*). Within vector species like mosquitoes, resistant genotypes are less likely to transmit pathogens, and this has the potential to reduce disease in vertebrate populations (*Beerntsen et al., 2000*). Where pathogens are contributing the decline of beneficial species like pollinators, high levels of genetic variation may allow populations to recover (*Maori et al., 2007*). Understanding the origins of genetic variation in susceptibility is therefore a fundamental question in infectious disease biology.

As pathogens are harmful, natural selection is expected to favour resistant host genotypes. Directional selection on standing genetic variation will drive alleles to fixation, removing variants from the

population (*Falconer, 1960*; *Falconer and Mackay, 1996*; *Blows and Hoffmann, 2005*). However, as directional selection also increases the frequency of new mutations that change the trait in the direction of selection, at equilibrium it is expected to have no effect on levels of standing genetic variation (relative to mutation-drift balance; *Hill, 1982*). However, selection mediated by pathogens may be different. Coevolution with pathogens can result in the maintenance of both resistant and susceptible alleles by negative frequency dependent selection (*Woolhouse et al., 2002*; *Haldane, 1949*). Similarly, when infection prevalence exhibits geographical or temporal variation, selection can maintain genetic variation, especially if pleiotropic costs to resistance provide an advantage to susceptible individuals when infection is rare (*Nuismer et al., 2003*; *Thompson, 1999*; *Koskella, 2018*). Even when there is simple directional selection on alleles that increase resistance, the direction of selection by pathogens may frequently change so populations may not be at equilibrium. If selection favours rare alleles – such as new mutations – directional selection can transiently increase genetic variation during their spread through the population (*Barton and Turelli, 1987*; *Bangham et al., 2007*; *Magwire et al., 2011*).

As part of a whole genome association study, we have previously estimated levels of genetic variation in the susceptibility of *D. melanogaster* to four different viruses (*Magwire et al., 2012*). We found that there was more genetic variation in susceptibility to the two viruses that were isolated from *D. melanogaster* than the two viruses from other insect species. Furthermore, in each of these naturally coevolved host-pathogen associations we detected a single major-effect polymorphism affecting resistance. This led us to propose that coevolution had increased genetic variation in susceptibility due to the presence of major-effect resistance polymorphisms. However, this conclusion remains anecdotal. First, aside from two sigma viruses, the viruses we used were mostly very distantly related, so their biology may differ for many reasons. Second, our association study had low statistical power, so conclusions about the genetics were based on just two genes. Finally, and most importantly, the link between coevolutionary history and genetic variation is based on a single host and four viruses, and so could arise by chance. For this reason, here we return to this question and formally test whether a history of coevolution alters the amount and nature of genetic variation.

To examine how selection by a pathogen affects levels of genetic variation we used a natural host-virus system; *Drosophila* and sigma viruses (*Longdon and Jiggins, 2012*; *Longdon et al., 2012*). Sigma viruses are a clade of insect RNA viruses with negative-sense genomes in the family Rhabdoviridae (*Longdon et al., 2017*; *Longdon et al., 2015a*; *Longdon et al., 2010*; *Longdon et al., 2011a*). They are vertically transmitted through eggs and sperm, and each sigma virus infects a single host species, simplifying studies of coevolution (*Longdon et al., 2017*; *Longdon et al., 2011a*). In *Drosophila melanogaster*, despite the virus causing little adult mortality, infection reduces host fitness by approximately 25% (*Wilfert and Jiggins, 2013*; *Yampolsky et al., 1999*). As prevalence in wild populations is typically around 10% (*Wilfert and Jiggins, 2013*), there is the necessary selective pressure for resistance to evolve (*Magwire et al., 2011*; *Cao et al., 2016*). In *D. melanogaster*, three major-effect resistance alleles have been identified (*Bangham et al., 2008a*; *Magwire et al., 2012*; *Bangham et al., 2007*; *Magwire et al., 2011*; *Cao et al., 2016*; *Bangham et al., 2008b*; *Wayne et al., 1996*). There has been a recent sweep of *D. melanogaster* sigma virus (DMelSV) genotypes that are able to overcome one of these host resistance genes (*Fleuriet, 1988*; *Wilfert and Jiggins, 2010a*; *Wilfert and Jiggins, 2014*). Given the power of *Drosophila* genetics, this system is an excellent model of a coevolutionary arms race between hosts and pathogens.

Sigma viruses offer a novel way to test how coevolution with a pathogen alters the amount of genetic variation in host susceptibility. As sigma viruses are vertically transmitted, we can be certain of which hosts and viruses are naturally coevolving and which are not. In this study we have used four species of *Drosophila* (*D. affinis, D. immigrans, D. melanogaster* and *D. obscura*) that shared a common ancestor approximately 40 million years ago (*Obbard et al., 2012*; *Tamura et al., 2004*), and their natural sigma viruses (DAffSV, DImmSV, DMelSV and DObsSV, which have amino acid identities of <55% in the most conserved gene) (*Longdon et al., 2015a*; *Longdon et al., 2010*; *Longdon et al., 2011b*). Despite their vertical mode of transmission, the phylogenies of the viruses and their hosts are incongruent, suggesting they have jumped between host species during their evolution (*Longdon et al., 2011b*). To test whether selection by viruses increases the amount of genetic variation in host susceptibility we have compared the viral load of endemic viruses that naturally infect each of the four host species to non-endemic viruses. We then examined how selection

by these viruses has altered the genetic architecture of resistance by mapping loci that confer resistance to endemic and non-endemic viruses in *D. melanogaster*.

## Results

### Genetic variation in susceptibility to infection is greatest in coevolved host-virus associations

To test whether selection by viruses has increased genetic variation in susceptibility to infection, we compared endemic host-virus associations with novel associations that have no history of coevolution. We used four different species of *Drosophila,* each of which is naturally host to a different sigma virus (*Longdon and Jiggins, 2012*; *Longdon et al., 2012*; *Longdon et al., 2017*; *Longdon et al., 2010*; *Longdon et al., 2011a*). We collected four species from the wild and created genetically diverse populations in the laboratory. Using flies from these populations we crossed single males to single females to create full-sib families. The progeny of these crosses were then injected with either the virus isolated from that species, or a virus isolated from one of the other species (*Figure 1*). Studies on DMelSV have shown that loci that reduce loads when the virus is injected also reduce infection rates in both the lab and field (*Bangham et al., 2008a*; *Longdon et al., 2012*; *Bregliano, 1970*; *Brun and Plus, 1980*; *Ohanessian-Guillemain, 1963*; *Wilfert and Jiggins, 2010b*). As infection is costly, this is expected to increase host fitness. Fifteen days post infection we extracted RNA from the flies and measured viral load by quantitative RT-PCR. The differences between the viral load of different families allowed us to estimate the genetic variance ($V_G$) in viral load – a measure of how much viral resistance varies in the population due to genetic as opposed to

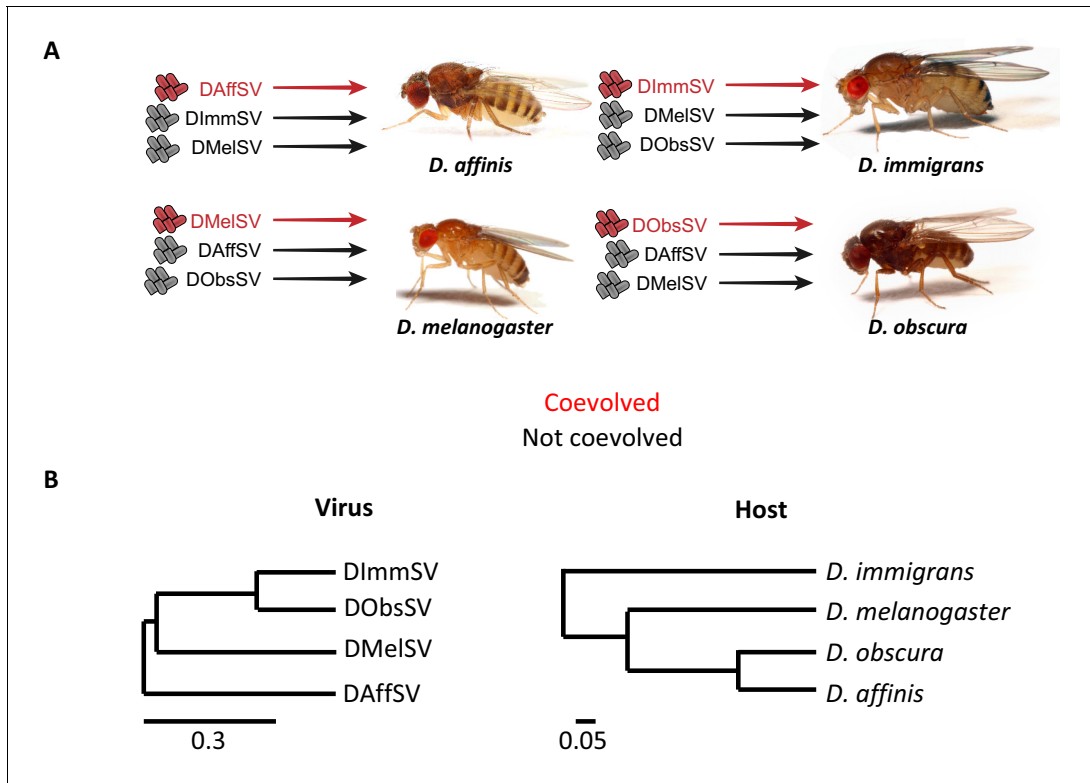

**Figure 1.** Experimental design and phylogenies. (A) Four species of *Drosophila* were independently infected both with a sigma virus with which they are naturally infected with in nature (red) and two viruses that naturally infect another species (black). (B) Phylogenies of the sigma viruses (inferred using the *L* gene) and their *Drosophila* hosts (inferred using *COI, COII, 28S rDNA, Adh, SOD, Amyrel* and *RpL32* genes), redrawn from *Longdon et al. (2015a)* and *Longdon et al. (2015b)*. Scale bars represent substitutions per site under a relaxed clock model. Posterior supports for nodes are all >0.99.
DOI: https://doi.org/10.7554/eLife.46440.002

environmental causes. In total we infected 52,592 flies and measured the viral load in 4295 biological replicates (a vial containing a mean of 12 flies) across 1436 full-sib families (details of sample sizes in Additional Methods).

Within populations of all four species, we found significantly greater genetic variance in susceptibility to the sigma virus that naturally infects that species compared to viruses from other species (*Figure 2*; *Figure 2—figure supplement 1*; *Supplementary file 1* and *2*). These different variances reflect considerable differences in the mean viral load between families (*Figure 2A*). For example, when families of *D. obscura* in the 2nd and 98th percentile were compared, there was a 1294 fold difference between the viral loads of the coevolved virus in the families (*Figure 2A*). In contrast, for the non-coevolved viruses there was a 27 fold difference in DMelSV loads and a 19 fold difference in DAffSV loads (see *Figure 2A* for statistics). The data in *Figure 2* is zero-centred to allow comparison of the variances, but across the four species, there was no consistent difference between the mean viral load in coevolved versus non-coevolved associations (i.e. the coevolved virus does not always replicate to higher levels suggesting this is not an artefact of simply replicating poorly in a host – see *Figure 2—figure supplement 1*). Additionally, there was no correlation between the genetic variance in viral load and the mean viral load (*Figure 2—figure supplement 1*, Spearman's correlation: $\rho$= -0.38, S=296, *P*=0.22).

## Major-effect genetic variants that are known to provide resistance to DMelSV do not protect against other viruses

To examine whether the genetic basis of resistance to coevolved and non-coevolved viruses was different we estimated the genetic correlations ($r_g$) in their viral loads. These represent the proportion of genetic variance in viral load between pairs of viruses that shares the same genetic causes. In *D. melanogaster* these were 0.40 for DMelSV-DAffSV, (95% CIs: 0.20, 0.61) and 0.25 for DMelSV-DObsSV (95% CIs: −0.01,0.47). Similar results were obtained using the *D. melanogaster* mapping population described below (*Supplementary file 3*). In the other species our estimates sometimes had wide credible intervals, but the genetic correlations between coevolved and non-coevolved viruses were mostly below 0.5 (*Supplementary file 3*; note correlations are frequently low between pairs of non-endemic viruses too). Therefore if natural selection increases genetic variation in susceptibility to a natural pathogen, there is expected to be a smaller effect on non-coevolved viruses.

In *D. melanogaster* a substantial proportion of the genetic variance in susceptibility to DMelSV is explained by major-effect variants in the genes *CHKov1* and *p62* (also known as *Ref(2)P*) (*Magwire et al., 2012*; *Bangham et al., 2007*; *Magwire et al., 2011*; *Wayne et al., 1996*; *Contamine et al., 1989*). The resistant allele in each of these genes has arisen recently by mutation and been driven up in frequency by natural selection, presumably due to the presence of DMelSV in natural populations (*Bangham et al., 2007*; *Magwire et al., 2011*). Therefore, if these genetic variants confer resistance to DMelSV but not the other sigma viruses, then this may explain the differences in genetic variance that we observed.

To examine whether *CHKov1* or *p62* contributed to the differences in genetic variance we observed in *D. melanogaster*, we genotyped the parents of the full sib families for variants that confer resistance (*Supplementary file 4*). Assuming the effects of the resistant alleles are additive, we estimated that the load of the coevolved virus DMelSV was more than halved in homozygous *CHKov1* resistant flies compared to susceptible flies (reduction in $\log_2$ viral load = 1.2, 95% CI = 0.6, 1.8). In contrast we found no significant effect of this gene on loads of the non-coevolved viruses (DAffSV = −0.2, 95% CI = 0.2,–0.8; DObsSV = −0.4, 95% CI = −0.04, 1.0). The resistant allele of *p62* was present at such a low frequency (1.5%) in the population that we lacked statistical power to investigate its effects.

To confirm these results we infected 1869 flies from 32 inbred *D. melanogaster* lines (*Mackay et al., 2012*) that had known *CHKov1* or *p62* genotypes. The effect of these genes was greater on the naturally occurring virus than the viruses from other species (*Figure 3*; effect of genotype on DMelSV load: $F_{2,28}$ = 13.2, p=0.00001; DAffSV: $F_{2,29}$ = 4.9, p=0.01; DObsSV: $F_{2,29}$ = 5.7, p=0.01).

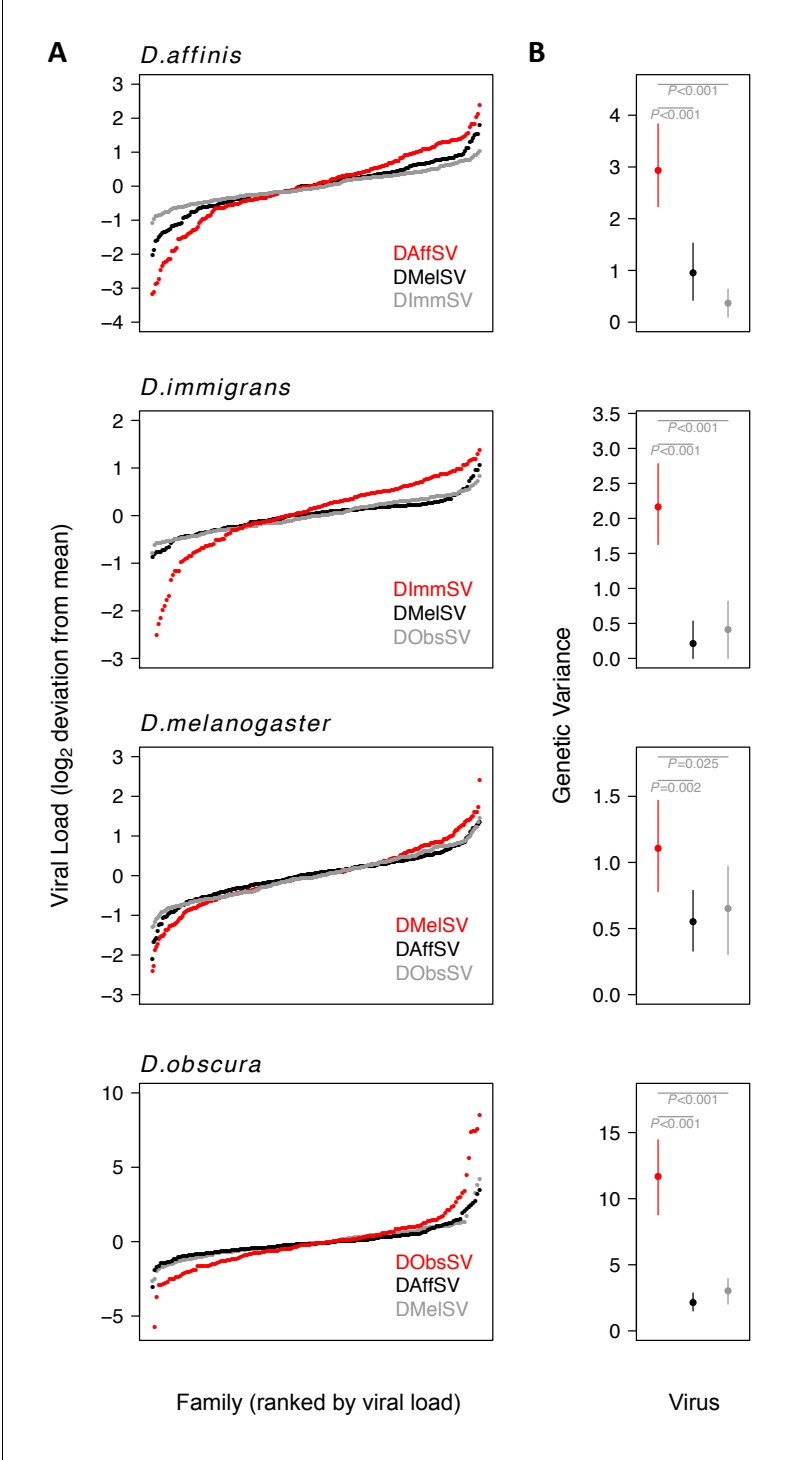

**Figure 2.** Genetic variation in susceptibility to coevolved and non-coevolved viruses. The viral load was measured 15 days post infection by quantitative RT-PCR relative to a *Drosophila* reference gene (*RpL32*). (**A**) The points show model prediction family means from our GLM and are centred on zero. The number of families in each panel was down-sampled so the same number of families is shown for each virus. Coevolved host-virus associations are in red. (**B**) The genetic variance in $\log_2$ viral load was estimated from the between family variance assuming that all genetic variance is additive. The bars are 95% credible intervals. Posterior probabilities for significantly different genetic variances are shown in grey (see *Supplementary file 1* and *2*).

DOI: https://doi.org/10.7554/eLife.46440.003

The following figure supplement is available for figure 2:

*Figure 2 continued on next page*

*Figure 2 continued*

**Figure supplement 1.** Estimates of genetic variance plotted against mean viral load for each species-virus combination.

DOI: https://doi.org/10.7554/eLife.46440.004

## There are a greater number of major-effect variants in coevolved host-virus associations

To investigate how coevolution shapes the genetics of resistance, we mapped loci controlling resistance using a *D. melanogaster* advanced intercross population (the DSPR panel [*King et al., 2012*]). This population samples genetic variation in a small number of genotypes from around the world (the experiments above sampled many genotypes from a single location). It was founded by allowing two sets of 8 inbred founder lines to interbreed for 50 generations, then creating recombinant inbred lines (RILs) whose genomes are a fine-scale mosaic of the original founder genomes. We used 377 RILs from these populations, which have up to 15 alleles of each gene (one founder line is shared between the two populations). We infected 15,916 flies across 1362 biological replicates (a vial containing a mean of 12 flies) with DMelSV, DAffSV or DObsSV and measured viral load as above (see Materials and methods).

We first estimated the genetic variance in viral load within our mapping population. The results recapitulated what we had found above in a natural population of flies — there was considerably more genetic variation in susceptibility to the coevolved virus than the non-coevolved viruses (*Figure 4A*, filled circles). Therefore, our earlier result from a single population holds when sampling

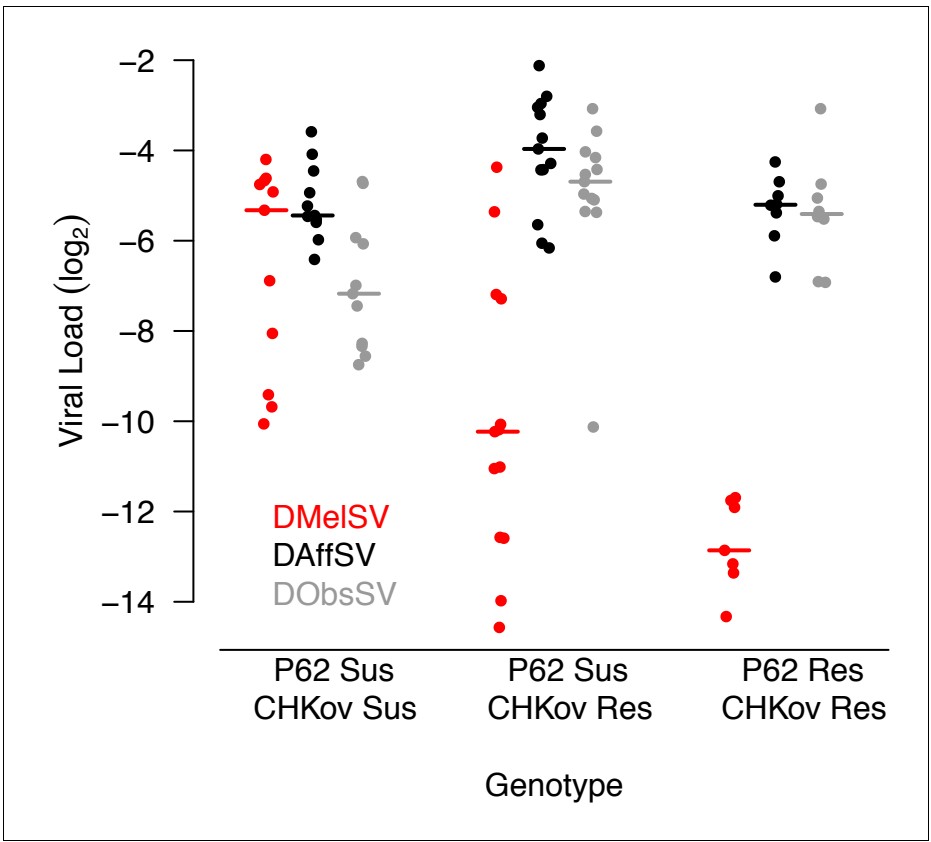

**Figure 3.** Viral load in *D. melanogaster* lines carrying different alleles of *CHKov1* and *p62*. Each point is the viral load of a separate inbred fly line carrying the resistant (Res) or susceptible (Sus) allele of *P62* or *CHKov1*. Horizontal bars are medians. Viral load was measured 15 days post infection by quantitative RT-PCR relative to a *Drosophila* reference gene (*RpL32*).

DOI: https://doi.org/10.7554/eLife.46440.005

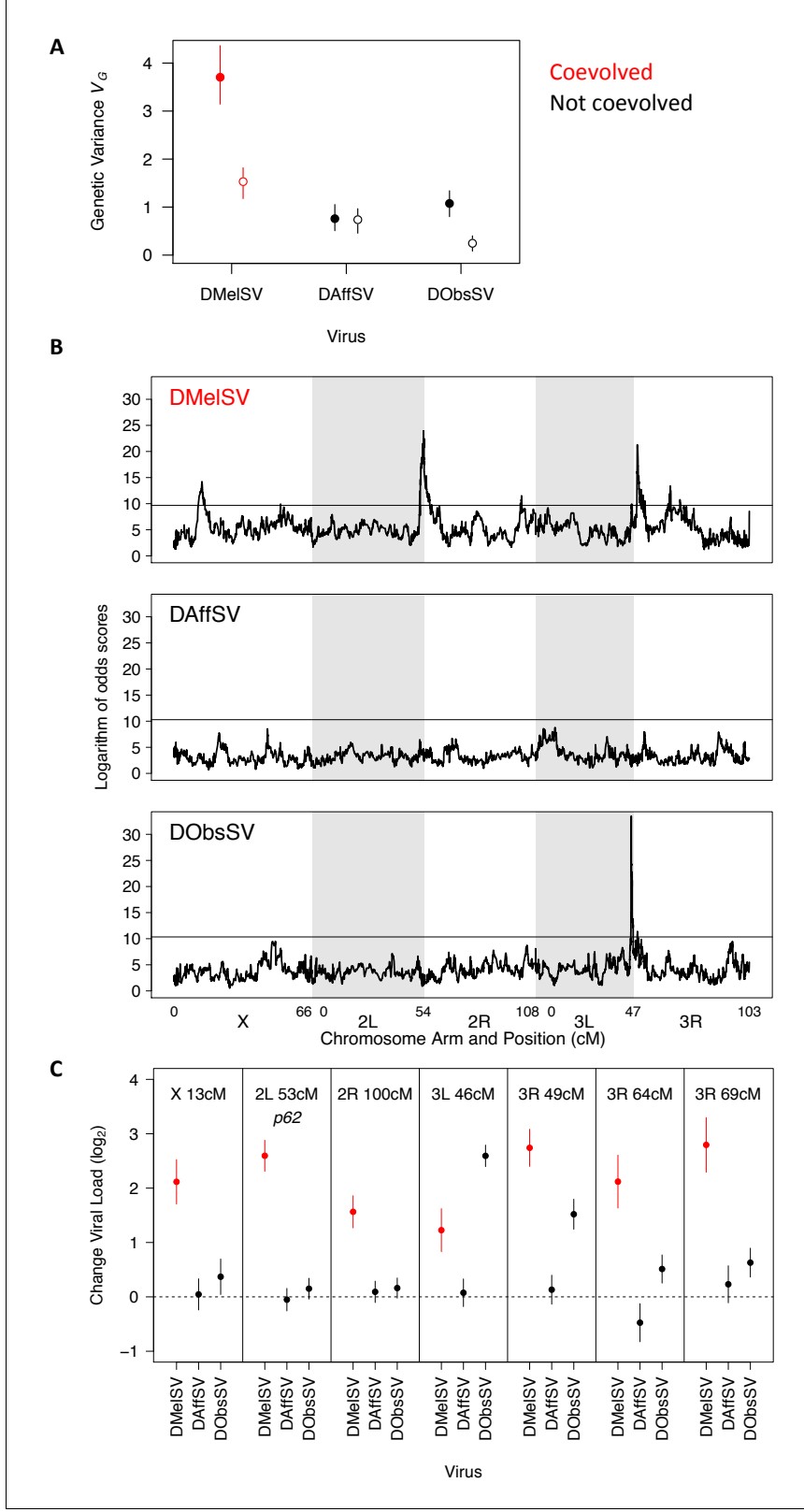

**Figure 4.** The genetic architecture of resistance to coevolved and non-coevolved viruses in *D. melanogaster*. (**A**) The genetic variance in viral load within the mapping population (filled circles). The open circles are estimates of the genetic variance after accounting for the effects of the QTL in panel C. Error bars are 95% credible intervals. (**B**) QTL affecting viral load. The horizontal line shows a genome-wide significance threshold of p<0.05 that was

*Figure 4 continued on next page*

*Figure 4 continued*
obtained by permutation of Logarithm of odds (LOD) scores. (**C**) The effect of the seven QTL detected on the load of the three viruses. Only QTL that remained were significant following multiple regression with all the loci are shown. The coevolved virus is shown in red.
DOI: https://doi.org/10.7554/eLife.46440.006

flies from across six continents, although the magnitude of the effect is considerably greater in this mapping population.

To examine the genetic basis of virus resistance, we looked for associations between genotype and viral load across the genome (*Figure 4B*). In the coevolved association (DMelSV) we identified seven QTL associated with resistance, compared to one that affects DObsSV and none affecting DAffSV (*Supplementary file 5*; this excludes one DMelSV QTL on the X chromosome that did not remain significant after accounting for the other QTL). The QTL affecting DObsSV also has a significant effect on DMelSV. One of the QTL corresponded to *p62* (2L 53 cM). The susceptible allele of *CHKov1* was not present in the fly lines assayed.

To examine the effect that the QTL have on viral load, we first split the founder alleles into a resistant class and a susceptible class (see Materials and methods) and then estimated the difference in viral load between the functionally distinct alleles. Six of the seven QTL resulted in greater reductions in the load of the coevolved virus (DMelSV) than the viruses isolated from other species (*Figure 4C*). There were only two cases where there was substantial cross-resistance to multiple viruses—3R 49 cM confers strong resistance to DMelSV and weak resistance to DObsSV, while 3L 46 cM confers weak resistance to DMelSV and strong resistance to DObsSV.

Together, this modest number of loci with substantial effects on resistance explains most of the high genetic variance in resistance to the coevolved virus (*Figure 4A*, filled versus open circles). Individually, resistant alleles cause an approximate 3–7 fold reduction in viral load (*Figure 4C*), and together they explain 59% of the genetic variance in susceptibility to DMelSV, 77% for DObsSV and 3% for DAffSV (*Figure 4A*, filled versus open circles). However, even after accounting for these genes there remains a significantly higher genetic variance in the viral load of the coevolved virus (*Figure 4A*, open circles, non-overlapping 95% CI).

## Discussion

We have found greater genetic variation in susceptibility to viruses that naturally infect *Drosophila* compared to viruses that do not, suggesting that selection by these pathogens has acted to increase the amount of genetic variation in susceptibility. This effect was largely caused by a modest number of major-effect genes that explain over half of the genetic variance in resistance.

As the genetic variants in the genes *p62 (ref(2)P)* and *CHKov1* that confer resistance to DMelSV have been identified, this has previously allowed us to use patterns of DNA sequence variation to infer how selection has acted on resistance in *D. melanogaster*. In both these genes the resistant alleles have arisen relatively recently by mutation and natural selection has pushed them rapidly up in frequency, leaving a characteristic signature of elevated linkage disequilibrium and low genetic diversity around the variant causing resistance (*Bangham et al., 2007*; *Magwire et al., 2011*; *Wayne et al., 1996*). There is no indication of negative frequency dependent selection, and these polymorphisms appear to have arisen from partial selective sweeps (*Bangham et al., 2007*; *Magwire et al., 2011*).

The most parsimonious explanation of these observations is that there has been directional selection favouring resistance alleles (although this type of data cannot rule out negative frequency dependent selection as is predicted by models of coevolution). At equilibrium, directional selection on a trait is not expected to affect its genetic variance (relative to a population under mutation-drift balance; *Hill, 1982*). However, the genetic variance will transiently increase if the variants under selection are initially at low frequency (*Barton and Turelli, 1987*), as was the case for both *p62* and *CHKov1* (*Bangham et al., 2007*; *Magwire et al., 2011*). A particular feature of pathogens is that the direction of selection is likely to continually change as new pathogens appear in populations or existing pathogens evolve to overcome host defences. For example, in France and Germany in the 1980s, DMelSV evolved to largely overcome the effects of the resistant allele of *p62* (*Fleuriet and*

*Periquet, 1993*; *Fleuriet and Sperlich, 1992*). Similarly, DImmSV has swept through European populations of *D. immigrans* in the last ~16 years and DObsSV through UK populations of *D. obscura* in the last ~11 years (*Longdon et al., 2017*; *Longdon et al., 2011a*). If selection by pathogens continually changes and resistance evolves from new mutations, then this may cause a sustained increase in genetic variance in susceptibility to infection.

A key question is whether the increased genetic variation that we see in coevolved *Drosophila–*sigma virus interactions will hold for coevolved pathogens more generally. Theory suggests that a critical factor determining levels of genetic variation is whether resistance is costly to evolve, as this can result in the maintenance of variation by negative frequency dependent selection (*Antonovics and Thrall, 1994*; *Boots et al., 2014*). In humans this has been proposed as an explanation of why there is less genetic variation in susceptibility to pathogens that are effectively controlled by the adaptive immune response, as these resistance mechanisms may be less costly (*Baker and Antonovics, 2012*). However, it seems unlikely that virus resistance in *Drosophila* is costly, as experiments have failed to detect costs of DCV resistance (*Faria et al., 2015*) despite costs of parasitoid and bacterial resistance being repeatedly detected (*McGonigle et al., 2017*; *McKean et al., 2008*; *Ye et al., 2009*). Sigma viruses are also extreme host specialists, so evolutionary changes in resistance will tend to alter pathogen prevalence and so the strength of selection. These epidemiological feedbacks are predicted to frequently increase genetic diversity (*Boots et al., 2014*; *Best et al., 2009*; *Boots and Haraguchi, 1999*). However, in *D. melanogaster* we see large amounts of genetic variation in susceptibility to viruses that have a broader host range than sigma viruses (DCV and Kallithea virus that infect *D. melanogaster* and *D. simulans* in the wild) (*Magwire et al., 2012*; *Palmer et al., 2018*; *Christian, 1987*; *Webster et al., 2015*). Therefore, it seems unlikely that our conclusions will be a quirk of the sigma virus system. In an analogous study of rust resistance in wild flax plants, sympatric (putatively coevolved) populations had fewer partial resistances than allopatric populations, suggesting more major gene effects, even though overall there was somewhat less genetic variation in susceptibility to sympatric fungal pathogens (*Antonovics et al., 2011*).

Quantitative traits are typically controlled by a very large number of genetic variants, each of which tends to have a very small effect (*Shi et al., 2016*; *Yang et al., 2010*). However, susceptibility to sigma viruses has a simpler genetic basis, with seven polymorphisms explaining over half the genetic variance. This confirms our previous work in *D. melanogaster* showing a simple genetic basis of virus resistance (*Magwire et al., 2012*; *Cogni et al., 2016*). As these genetic variants mostly only affect the naturally occurring pathogen of *D. melanogaster,* our results suggest that not only is selection by pathogens increasing the genetic variance but it is also altering the genetic architecture of resistance by introducing major-effect variants into the population. One explanation for this observation is that most quantitative traits are under stabilising selection, so major effect variants will tend to be deleterious and removed by selection (*Gibson, 2018*). In contrast, selection by pathogens likely changes through time and populations may be far from their optimal level of resistance. If this is the case, Fisher's geometric model predicts that major effect variants will be favoured by directional selection (*Fisher, 1930*). Alternatively, the coevolution of hosts and parasites can favour discrete susceptible and resistant hosts (*Boots et al., 2014*), and at the genetic level this may result in major-effect variants (although this theory does not explicitly address the underlying genetics). The simple genetics may also be driven by mutation—for many traits major-effect mutations that increase fitness may be extremely rare, but this may not be the case for virus resistance. For example a single (loss of function) mutation may prevent a virus binding to a host receptor or utilising other parts of the host cellular machinery, and so confer strong resistance.

Regardless of its causes, it may be common that susceptibility to infectious disease has a simple genetic basis. In humans, *Hill (2012)* advocated the view that susceptibility to infectious disease is qualitatively different from other traits and has a much simpler genetic basis (*Hill, 2012*). In *Drosophila*, resistance to DCV and parasitoid wasps both have a simple genetic basis (*Magwire et al., 2012*; *Cao et al., 2017*; *Orr and Irving, 1997*). In plants, major-effect polymorphisms in R genes are commonplace (*Hammond-Kosack and Jones, 1997*) and in a plant-fungi system genotype-by-genotype interactions explain a larger proportion of the total variance in sympatric (more coevolved) associations (*Antonovics et al., 2011*). In contrast, studies of bacterial resistance in *Drosophila* have typically used pathogens that are unlikely to have any history of coevolution, and have found a polygenic basis to resistance (*Bou Sleiman et al., 2015*; *Hotson and Schneider, 2015*; *Howick and Lazzaro, 2017*; *Wang et al., 2017*). In these studies the polymorphism with the largest effect was

found against the only natural *D. melanogaster* pathogen tested (a polymorphism in *Diptericin* detected using *Providencia rettgeri* infection), anecdotally supporting the patterns seen here (*Howick and Lazzaro, 2017*; *Unckless et al., 2015*).

A major source of emerging infectious disease is pathogens jumping into novel hosts where they have no co-evolutionary history (*Longdon et al., 2014*; *Parrish et al., 2008*). Our results suggest that when a pathogen infects a novel host species, there may be far less genetic variation in suscep- tibility among individuals than is normally the case. This may create a 'monoculture effect' (*King and Lively, 2012*; *Lively, 2010*; *Ostfeld and Keesing, 2012*), which could leave populations vulnerable to epidemics of pathogens that have previously circulated in other host species. Longer term, low levels of pre-standing genetic variation may slow down the rate at which the new host can evolve resistance to a new pathogen.

In conclusion, we have demonstrated that selection by pathogens has increased the amount of genetic variation in host susceptibility. We find resistance has a simple underlying genetic architec- ture and is largely controlled by major effect resistance loci.

## Materials and methods

### Virus extraction and infection

We extracted the sigma viruses DAffSV, DImmSV, DMelSV and DObsSV from infected stocks of *D. affinis* (line: NC10), *D. immigrans* (line: DA2), *D. melanogaster* (line: E320 Ex) and *D. obscura* (line: 10A) respectively (*Longdon et al., 2017*; *Longdon et al., 2015a*; *Longdon et al., 2010*; *Longdon et al., 2011a*; *Brun and Plus, 1980*). These infected lines were collected from the wild between 2007–2012 (all from the UK, bar *D.affinis* which was collected in the USA). Infected fly stocks were checked for infection by exposing the flies to 100% $CO_2$ at 12°C for 15mins then para- lysed flies were collected 30mins later. The DImmSV infected line does not show $CO_2$ sensitivity and so was confirmed to have a high level of infection using RT-PCR. Infected flies were frozen at −80°C and later homogenised in ringers solution (2.5 μl per fly) and centrifuged at 13,000 g for 10 min at 4° C. The supernatant was collected, 2% v/v FBS was added then virus solutions were aliquoted and stored at −80°C.

### Experimental design

We set up a common garden experiment to measure genetic variation in susceptibility to natural and non-natural viruses across four host species. In a fractional factorial experiment each species was infected it with its own virus, as well as two viruses that do not infect that host species (see *Fig- ure 1*). All of the viruses replicate in all hosts (with the exception of DImmSV in *D. melanogaster* that showed limited evidence of replication – this combination was not used in the experiment). All fly stocks used (see additional methods for stock details) were tested for existing sigma virus infection using RT-PCR over two generations. For all species we collected flies from the wild and we used a full-sib mating design. The progeny of these crosses were infected by injecting them with 69 nl of the viruses intrathoracically and measuring viral RNA loads 15 days post infection, as in *Longdon et al. (2011c)*. This time point was selected as RNA viral load tends to plateau from around day 15 post infection and there is no mortality from infection in this period. The specifics for each host species, including sample sizes, are detailed in the additional methods.

### Known resistance genes in *D. melanogaster*

We genotyped parents of each *D. melanogaster* full sib family from the experiment above for two resistance alleles that are known to confer protection against DMelSV; *p62 (Ref(2)P)* and *CHKov1*. We genotyped parental flies using PCR assays that produce different sized products depending on whether flies carry resistant or susceptible alleles. Information on these PCRs and primer sequences can be found in *Supplementary file 4*. We then calculated the number of resistance alleles in each family by summing the number of alleles from both mothers and fathers. We produced genotype information for 230 of the 255 families.

Another resistance allele has been identified in the gene *Ge-1* (*Cao et al., 2016*). However, this allele has been found to occur at a low frequency in wild populations. We genotyped 184 parental

flies from our experiment (parental flies for some families could not be collected) and found the resistant allele was not present, suggesting it is rare or absent.

We further examined the effect of alleles known to affect susceptibility to DMelSV on all three viruses. Firstly, we infected 32 lines from the Drosophila Genetic Reference Panel (DGRP) (*Mackay et al., 2012*) that were susceptible for both *p62* and *CHKov1* ($n$ = 11 lines), were resistant for *CHKov1* only ($n$ = 13 lines), or were resistant for both genes ($n$ = 8 lines) with DMelSV, DAffSV and DObsSV. No lines in the panel were resistant for *p62* and susceptible for *CHKov1*. We infected a mean of 18 flies per line (range = 3–22).

## Mapping resistance genes in *D. melanogaster*

We used 377 DSPR lines (154 from panel A and 223 from panel B, http://FlyRILs.org [*King et al., 2012*; *Long et al., 2014*]), kindly provided by S.J. Macdonald, University of Kansas) to carry out a Quantitative Trait Locus (QTL) study to examine the genetic basis of resistance to DAffSV, DMelSV and DObsSV in *D. melanogaster*.

Three females and three males from each DSPR line were placed into yeasted cornmeal vials and allowed to lay for 3–4 days, at 25˚C. Male offspring were collected at 0–4 days post-eclosion, and placed at 18˚C for 4–6 days. Flies were then injected with DMelSV, DAffSV or DObsSV as described above. Injected males were maintained on unyeasted cornmeal at 18˚C, and frozen on day 15 post-infection as above.

Injections were carried out over 13 weeks. Each day of injection a mean of 47 unique lines (range 20–60) and 51 replicate vials were injected with 1–3 different viruses. In total we assayed 377 DSPR lines (108 lines had two biological replicates). Each replicate vial contained a mean of 12 flies (range 1–22). In total, 15,916 flies were injected across both panels of DSPR lines. We injected 319 lines with all three viruses, 38 with 2 viruses and 20 with one virus. The order of injection of lines and of viruses, was randomised across injection days. Independent biological replicates were injected on different days. Panel A and Panel B lines were assayed in two overlapping blocks.

## Measuring viral load

We measured the change in RNA viral load using qRT-PCR. The viral RNA load was expressed relative to the endogenous control housekeeping gene *RpL32* (*Rp49*). RNA was extracted from flies homogenised in Trizol and reverse transcribed with GoScript reverse transcriptase (Promega) and random hexamer primers, and then diluted 1:10 with nuclease free water. The qRT-PCR was performed on an Applied Biosystems StepOnePlus system using Sensifast Hi-Rox Sybr kit (Bioline) with the following PCR cycle: 95˚C for 2 min followed by 40 cycles of: 95˚C for 5 s followed by 60˚C for 30 s. Two qRT-PCR reactions (technical replicates) were carried out per sample with both the viral and endogenous control primers. Each qRT-PCR plate contained three standard samples, and all experimental samples were split across plates in a blocked design. A linear model was used to correct the cycle threshold (Ct) values for differences between qRT-PCR plates. Primer sequences are in *Supplementary file 6*.

To estimate viral load, we calculated ΔCt as the difference between the qRT-PCR cycle thresholds of the virus and the endogenous control. Viral load calculated without using the endogenous control is strongly correlated to ΔCt for all species.

## Statistical analysis full-sib experiments

We used a linear mixed model to examine the amount of genetic variation in susceptibility to the different viruses. We used a trivariate model with the load of the three viruses as the response variable. For each species the model was structured as:

$$y_{vfi} = \beta_{1:v} + u_{v:f} + u_{v:d} + e_{vfi} \tag{1}$$

Where $y_{vfi}$ is the $\log_2$ viral load of the $i^{th}$ biological replicate of full-sib family $f$ infected with virus $v$. $\beta$ are the fixed effects, with $\beta_1$ being the mean viral load of each virus. $u$ are the random effects for full-sib families ($f$) and for the day of injection ($d$), $e$ are the residuals. By assuming that all the genetic variation in the population is additive (*Hill et al., 2008*), we estimated the genetic variance ($V_G$) of the viral load as twice the between-family variance (*Falconer, 1960*; *Falconer and Mackay,*

*1996*). Both empirical data and theory suggest additive genetic variation makes up large proportion of the total genetic variance (*Hill et al., 2008*).

In addition, for *D. melanogaster* we ran a further model that included the additional fixed effects $\beta_{2:v}$ and $\beta_{3:v}$ that are the linear effects of the *CHKov1* and *p62* (*Ref(2)P*) resistance alleles. We assumed these genes had additive effects, and modelled their effects simply as the proportion of resistant alleles in a family (if one parent was heterozygous and the other homozygous susceptible, the value is 0.25).

The model was fitted using the MCMCglmm package in R (*Hadfield, 2010*). The random effects (and residuals) are assumed to be multivariate normal with zero mean and covariance structure $\mathbf{V} \otimes \mathbf{I}$. $\mathbf{I}$ is an identity matrix, and $\mathbf{V}$ a matrix of estimated variances and covariances. For the random effects $\mathbf{V}$ is a $3 \times 3$ covariance matrix describing the variances for each virus and the covariances between them. The off-diagonal elements of $\mathbf{V}$ for the residual were set to zero because the covariances between traits at these levels are not estimable by design.

Diffuse independent normal priors were placed on the fixed effects (means of zero and variances of $10^8$). Parameter expanded priors were placed on the covariance matrices resulting in scaled multivariate $F$ distributions which have the property that the marginal distributions for the variances are scaled (by 1000) $F_{1,\ 1}$. The exceptions were the residual variances for which an inverse-gamma prior was used with shape and scale equal to 0.001. The MCMC chain was ran for 130 million iterations with a burn-in of 30 million iterations and a thinning interval of 100,000.

We confirmed the results were not sensitive to the choice of prior by also fitting models with inverse-Wishart and flat priors for the variance covariance matrices (described in *Longdon et al., 2011c*), as well as fitting the models by REML in ASReml R (*Gilmour et al., 2002*). These analyses all gave qualitatively similar results (data not shown).

## Statistical analysis of QTL experiment

Based on genotyping data, the probability that each Recombinant Inbred Line (RIL) in the DSPR panel was derived from each of the eight founder lines has been estimated at 10 kB intervals across the genome (*King et al., 2012*). To identify QTL affecting viral load we first calculated the mean viral load ($\Delta Ct$) across the biological replicates of each RIL. We then regressed the mean viral load against the eight genotype probabilities and calculated logarithm of odds (LOD) scores using the DSPRqtl package in R (*King et al., 2012*). These LOD scores were calculated separately for DSPR Panel A and Panel B, and then summed at each genomic location. To obtain a significance threshold, we permuted our mean viral load estimates across the RILs within each panel, repeated the analysis above and recorded the highest LOD score across the entire genome. This process was repeated 1000 times to obtain a null distribution of the maximum LOD score.

To estimate the effect of each QTL we assumed that there was a single genetic variant affecting viral load, so the founder alleles could be assigned to two functionally distinct allelic classes. First, we regressed the mean viral load against the genotype probabilities (as described above), resulting in estimates of the mean viral load of each founder allele in the dataset. The two DSPR panels had one founder line (line 8) in common. For QTL where the line eight allele was present in both panels, this analysis included data from both panels, and 'panel' was included as a fixed effect in the analysis. When this was not the case, we analysed only data from the panel where the QTL was most significant. We then ranked the founder alleles by viral load estimate, and split this ranked list into all possible groups of two alleles. For each split, the genotype probabilities in the first group of founder alleles were summed. We then regressed mean viral load against each of these combined genotype probabilities. The regression model with the highest likelihood was taken as the most likely classification into allelic classes. The effect size of the QTL was then estimated from this model.

To estimate the genetic variance in viral load within the DSPR panels we modified the model described in *Equation 1* as follows. $y_{vfi}$ is the $\log_2$ viral load of the $i^{th}$ biological replicate of each RIL $f$ infected with virus $v$. There was a single fixed effect, $\beta$, of the panel the line is from. $u$ is the random effect for each RIL ($f$). The day of injection ($d$) was omitted. As all the RILs are homozygous, we estimated the genetic variance in viral load ($V_G$) as half the between-RIL variance. This assumes all the genetic variation is additive.

To estimate the proportion of the genetic variance that is explained by the QTL we identified, we repeated this analysis but included the 7 QTL we identified as fixed effects in the model. Each QTL was included by estimating the probability that each line carried the resistant allele of the QTL and

adding this as a fixed effect to the model. The between-RIL variance then allowed us to estimate the genetic variance in viral load after removing the effects of the QTL.

## Additional methods

All lines were screened for their retrospective sigma virus over two generations by RT-PCR, and infected isofemale lines discarded prior to the experiment.

### *Drosophila melanogaster*

We created an outcrossed population by combining 150 isofemale lines of *D. melanogaster* (collected in Accra, Ghana (5.593,–0.188) in 2014) in a population cage. The population was maintained throughout the experiment with a large population size (~1500–2000 flies), with eggs collected from the population cage used to set up each subsequent generation. All rearing was carried out on cornmeal medium (recipe below) sprinkled with live yeast ('yeasted') at 25℃.

Virgin flies were collected daily from bottles set up at a controlled egg density. Full-sib families were set up using crosses of single male and female virgins placed in the same vial and aged for 3 days. Each of these families was tipped onto fresh food daily for 5 days to create replicate vials. After 5 days the adult flies were frozen for later genotyping. 12 days after laying, male offspring were collected from each replicate vial and split into two vials of cornmeal medium without any yeast on the surface ('unyeasted') and placed at 18℃.

After 5 days these flies were injected with 69 nl of virus extract intra-abdominally using a Nanoject II micro-injector (Drummond scientific). Injected flies were kept at 18℃ and tipped onto fresh unyeasted cornmeal every 5 days, before being homogenised in Trizol (Invitrogen) and frozen at −80℃ on day 15 post injection for later RNA extraction and qRT-PCR.

Injections were carried out over 25 overlapping blocks. Each block consisted of 10 families, and each day, two vials per family were injected with two different viruses. Each replicate vial contained a mean of 14 flies (range 3–28 flies). In total we measured 255 families over 1567 biological replicates. We aimed to carry out a minimum of 2 replicates of each virus per family, but where possible we carried out 3 or four replicates (92 virus-family combinations had one replicate, 555 had two replicates, 82 had 3 replicates and 25 had four replicates.). 248 families had replicates for all three viruses. Blocks were staggered to overlap with at least 20 families being infected on any one day. The order families were injected in was randomised, and the order the different viruses were injected was blocked across days.

### *D. immigrans*

92 *D. immigrans* lines were collected from Madingley, Cambridge, UK (52.225, 0.043) in 2012 and 2015. Full-sib families were set up using crosses between the 92 isofemale lines of *Drosophila immigrans*. Flies were reared on malt food (recipe below) at 18℃. Crosses were between different isofemale lines (i.e. excluding reciprocal crosses) and maximising the number of lines used. Families were established from 2 to 4 day old single female and male virgin flies placed in the same vial for 7 days. These crosses were tipped onto fresh food every 7 days to generate replicate vials of each family. Eclosed males were collected 27–34 days after initial egg laying and injected with DImmSV, DMelSV or DObsSV 1–3 days post-collection, then maintained and frozen on day 15 post-infection as above.

Injections were carried out over 18 overlapping blocks. Each block consisted of an average of 19 families and 46 replicate vials. Each replicate vial contained a mean of 14 flies (range: 4–26). In total we assayed 341 families over 812 biological replicates. We aimed to have a minimum of 2 replicates per virus per family (235 virus-family combinations had one replicate, 270 had two replicates, 11 had 3 replicates and 1 had four replicates). 140 families had replicates across two different viruses and 18 families had replicates for all three viruses. Blocks were staggered to overlap with a mean of 39 families being infected on any one day. The order families were injected in was randomised, and the order the different viruses were injected was blocked across days.

## D. affinis

| Site | N |
| --- | --- |
| Athens, Georgia, USA, (33.946,–83.384) in 2012 | 13 |
| Great Smokey Mountain National Park, Gatlinburg, USA (35.698,–83.613) in 2015 | 23 |
| Rochester, New York, USA (43.135,–77.599) in 2012 | 4 |

Full-sib families were set up using crosses between 40 isofemale lines of *Drosophila affinis* (see above for collection details) collected in the U.S. Flies were reared on malt food (recipe below) at 18°C. Crosses were between different isofemale lines (i.e. excluding reciprocal crosses) and maximising the number of lines used. Families were established from 6 day old single female and male virgin flies placed in the same vial for 7 days. These crosses were tipped onto fresh food every 7 days to generate replicate vials of each family. Eclosed males were collected 35–42 days after initial egg laying and then injected with DAffSV, DImmSV or DMelSV 1–3 days post-collection, then maintained and frozen on day 15 post-infection as above.

Injections were carried out over 27 overlapping blocks. Each block consisted of an average of 19 families and 28 replicate vials. Each replicate vial contained a mean of 11 flies (range: 3–23). In total we assayed 520 families over 1003 biological replicates. We aimed to have a minimum of 2 replicates per virus per family (336 virus-family combinations had one replicate, 286 had 2 replicates and 30 had three replicates). 109 families had replicates across two different viruses and 12 families had replicates for all three viruses. Blocks were staggered to overlap with a mean of 23 families being infected on any one day. The order families were injected in was randomised, and the order the different viruses were injected was blocked across days.

## D. obscura

| Site | N |
| --- | --- |
| Derbyshire Site A, UK (52.978,–1.440) in 2012 | 4 |
| Derbyshire Site C, UK (52.903,–1.374), in 2012 | 1 |
| Les Gorges du Chambon, France (45.622, 0.555) in 2012 | 1 |
| Madingley, Cambridge, UK (52.226, 0.046) in 2014 | 15 |

*D. obscura* were collected in the United Kingdom and France (see above). Males and females were separated, and females were placed in vials to establish isofemale lines. Full-sib families were set up using crosses between 21 isofemale lines of *Drosophila obscura* collected in the UK. Flies were reared on banana food (recipe below) at 18°C. Crosses were between different isofemale lines (i.e. excluding reciprocal crosses) and maximising the number of lines used. Families were established from 6 day old single female and male virgin flies placed in the same vial for 7 days. These crosses were tipped onto fresh food every 7 days to generate replicate vials of each family. Eclosed males were collected 35–42 days after initial egg laying and then injected with DAffSV, DMelSV, or DObsSV 1–3 days post-collection, then maintained and frozen on day 15 post-infection as above.

Injections were carried out over 25 overlapping blocks. Each block consisted of a mean of 16 families with a mean of 76 vials being injected each day for 12 days. Each replicate vial contained a mean of 8 flies (range: 1–15). In total we assayed 320 families over 913 biological replicates. We aimed to have a minimum of 2 replicates per virus per family (126 virus-family combinations had one replicate, 314 had 2 replicates and 49 had 3 replicates and 3 had four replicates). 94 families had replicates across two different viruses and 39 families had replicates for all three viruses. Blocks were staggered to overlap with at least 10 families being infected on any one day. The order families were injected in was randomised, and the order the different viruses were injected was blocked across days.

## Sample size estimation

The number of full-sib families required for estimating genetic variance in susceptibility was determined by simulation using previous estimates of genetic variation to DMelSV in *D. melanogaster* (*Magwire et al., 2012*). After carrying out the full-sib experiment in *D. melanogaster*, we then down-sampled this data to calculate the minimum number of families required to provide accurate estimates for the other species. Sample sizes for the DSPR experiment were based on previous data (*Faria et al., 2015*).

## Food recipes

Banana:

    Mixture 1
        1000 ml water
        30 g yeast
        10 g agar
    Mixture 2
        20 ml Nipagin
        150 g pureed banana
        50 g corn syrup
        30 g malt powder

Bring mixture one to the boil for 3–4 min, whisk constantly. Add to Mixture two to Mixture 1. Whisk constantly and simmer for 5 min.

Cornmeal:

1200 ml water
13 g agar
105 g dextrose
105 g maize
23 g yeast

Combine and bring to a boil for 5mins, cool to 70°C before adding 35 ml Nipagin (10%)

Malt:

1000 ml water
10 g agar
60 g semolina
20 g yeast
80 g malt extract

Combine and bring to a boil for 5mins, cool to 70°C and then add 14 ml Nipagin (10%) and 5 ml propionic acid.

## Data availability

Datasets and R code for estimating the amount of genetic variation in susceptibility https://doi.org/10.6084/m9.figshare.6743339
    DGRP dataset https://doi.org/10.6084/m9.figshare.6743354
    DSPR dataset and R code https://doi.org/10.6084/m9.figshare.7195751

## Acknowledgements

Many thanks to: Alastair Wilson and Jarrod Hadfield for useful advice and discussion; Stuart Macdonald for providing DSPR lines, Trudy Mackay for providing the DGRP fly lines and Kelly Dyer with help collecting *D. affinis*; Darren Obbard for providing photographs of *Drosophila* species; Camille Bonneaud, Katherine Roberts, Ryan Imrie and the Unckless lab group for constructive comments on this work. Many thanks to Brian Lazzaro, Janis Antonovics, Bruno Lemaître and one anonymous reviewer for constructive comments that greatly improved the manuscript.

## Additional information

### Funding

| Funder | Grant reference number | Author |
|---|---|---|
| NERC Environmental Bioinformatics Centre | NE/L004232/1 | Elizabeth ML Duxbury<br>Jonathan P Day<br>Francis M Jiggins<br>Ben Longdon |
| Wellcome | Sir Henry Dale Fellowship (Grant Number 109356/Z/15/Z) | Ben Longdon |
| H2020 European Research Council | 281668 | Elizabeth ML Duxbury |
| Royal Society | Sir Henry Dale Fellowship (Grant Number 109356/Z/15/Z) | Ben Longdon |

The funders had no role in study design, data collection and interpretation, or the decision to submit the work for publication.

### Author contributions

Elizabeth ML Duxbury, Data curation, Formal analysis, Supervision, Investigation, Writing—original draft, Writing—review and editing; Jonathan P Day, Data curation, Formal analysis, Validation, Investigation, Visualization, Methodology, Writing—original draft, Project administration, Writing—review and editing; Davide Maria Vespasiani, Yannik Thüringer, Ignacio Tolosana, Sophia CL Smith, Lucia Tagliaferri, Altug Kamacioglu, Imogen Lindsley, Luca Love, Investigation, Writing—original draft, Writing—review and editing; Robert L Unckless, Resources, Writing—review and editing; Francis M Jiggins, Ben Longdon, Conceptualization, Resources, Data curation, Formal analysis, Supervision, Funding acquisition, Validation, Investigation, Visualization, Methodology, Writing—original draft, Project administration, Writing—review and editing

### Author ORCIDs

Elizabeth ML Duxbury (iD) http://orcid.org/0000-0002-5733-3645
Ignacio Tolosana (iD) http://orcid.org/0000-0002-3766-8296
Robert L Unckless (iD) http://orcid.org/0000-0001-8586-7137
Francis M Jiggins (iD) https://orcid.org/0000-0001-7470-8157
Ben Longdon (iD) http://orcid.org/0000-0001-6936-1697

### Decision letter and Author response

Decision letter https://doi.org/10.7554/eLife.46440.021
Author response https://doi.org/10.7554/eLife.46440.022

## Additional files

### Supplementary files

• Supplementary file 1. Credible intervals of the differences between estimates of genetic variance in susceptibility across host species and viruses. The natural virus for each host is in red and bold. Genetic variances are estimated from the among-family variances in viral load. 95% CIs show differences in estimates of genetic variation for different host-virus combinations, intervals that do not cross zero represent statistically significant differences.
DOI: https://doi.org/10.7554/eLife.46440.007

• Supplementary file 2. Estimates of genetic variation for each host virus combination. The natural virus for each host is in red and bold. Genetic variances are estimated from the among-family variances in viral load.
DOI: https://doi.org/10.7554/eLife.46440.008

• Supplementary file 3. Table S3 Genetic correlations ($r_g$) in viral loads across host species after infection by coevolved and non-coevolved viruses. The coevolved (natural) virus for each host is in red and bold. Models ran using REML gave similar estimates.
DOI: https://doi.org/10.7554/eLife.46440.009

• Supplementary file 4. Table S4 Primers for genotyping *D. melanogaster* resistance genes *Ge-1, p62 (Ref(2)P)* and *CHKov-1*. PCRs were carried out using a touchdown PCR cycle (95℃ 30sec, 62℃ (-1℃ per cycle) 30sec, 72℃ 1min; for 10x cycles followed by; 95℃ 30sec, 52℃ 30sec, 72℃ 1min; for a further 25x cycles).
DOI: https://doi.org/10.7554/eLife.46440.010

• Supplementary file 5. Table S5 QTL and their locations.
DOI: https://doi.org/10.7554/eLife.46440.011

• Supplementary file 6. Table S6 Primers for qRT-PCR (5'-3'). RpL32 primers overlap an intron-exon boundary. Sigma virus primers cross gene boundaries except for DImmSV that amplifies the L gene
DOI: https://doi.org/10.7554/eLife.46440.012

• Transparent reporting form
DOI: https://doi.org/10.7554/eLife.46440.013

## Data availability

All data generated or analysed during this study are available at Figshare: Datasets and R code for estimating the amount of genetic variation in susceptibility https://doi.org/10.6084/m9.figshare.6743339; DGRP dataset https://doi.org/10.6084/m9.figshare.6743354; DSPR dataset and R code https://doi.org/10.6084/m9.figshare.7195751.

The following datasets were generated:

| Author(s) | Year | Dataset title | Dataset URL | Database and Identifier |
|---|---|---|---|---|
| Elizabeth ML Duxbury, Jonathan P Day, Davide Maria Vespasiani, Yannik Thüringer, Ignacio Tolosana, Sophia CL Smith, Lucia Tagliaferri, Altug Kamacioglu, Imogen Lindsley, Luca Love, Robert L Unckless, Francis M Jiggins, Ben Longdon | 2019 | Susceptibility of different Drosophila melanogaster DGRP lines to three sigma viruses | https://doi.org/10.6084/m9.figshare.6743354 | Figshare, 10.6084/m9.figshare.6743354 |
| Elizabeth ML Duxbury, Jonathan P Day, Davide Maria Vespasiani, Yannik Thüringer, Ignacio Tolosana, Sophia CL Smith, Lucia Tagliaferri, Altug Kamacioglu, Imogen Lindsley, Luca Love, Robert L Unckless, Francis M Jiggins, Ben Longdon | 2019 | Natural selection by pathogens increases genetic variation in susceptibility to infection | https://doi.org/10.6084/m9.figshare.6743339 | Figshare, 10.6084/m9.figshare.6743339 |
| Elizabeth ML Duxbury | 2019 | Susceptibility of Drosophila melanogaster DSPR lines to three sigma viruses | https://doi.org/10.6084/m9.figshare.7195751 | Figshare, 10.6084/m9.figshare.7195751 |

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
