## [Decision Letter]

[Editors’ note: a previous version of this study was rejected after peer review, but the authors submitted for reconsideration. The first decision letter after peer review is shown below.]

Thank you for submitting your work entitled "Host-pathogen coevolution increases genetic variation in susceptibility to infection" for consideration by *eLife*. Your article has been reviewed by three peer reviewers, and the evaluation has been overseen by a Reviewing Editor and a Senior Editor. The following individuals involved in review of your submission have agreed to reveal their identity: Brian P Lazzaro (Reviewer #1); Janis Antonovics (Reviewer #3).

Our decision has been reached after consultation between the reviewers. Based on these discussions and the individual reviews below, we regret to inform you that your work will not be considered further for publication in *eLife*.

As you will note, reviewer's comments were generally positive about your manuscript. Nevertheless, in the course of the discussion between reviewers, two issues arise that justify the decision:

1) The present paper was judged too similar to a previous study of the authors (see reviewer 2). It was felt that the authors have already shown this result with *melanogaster* and that the present study is expanding on that previous observation using more host species. Thus the novelty was questioned.

2) The generality of the result might be overstated. Reviewers felt that the authors over-claim the implication for general host-parasite evolution, and that the pattern they described might only apply to infections for which resistance has a simple genetic architecture and the pathogen is rapidly evolving, although that may capture most specialist viruses.

Based on these concerns, it was decided to reject the manuscript but allow re-submission. At this stage, you are allowed to re-submit a revised manuscript that will be treated as a new submission and try to convince the reviewers of the importance of your manuscript.

*Reviewer #1:*

Duxbury, Day and colleagues present a clever test of whether specialist coevolution between a virus and its host increases the genetic variance for resistance in the host population. The paper is conceptually and methodologically sound. It is interesting and well written and could be published without modification. I offer only a few suggestions for improvement, and these are largely discretionary.

I have one major point of interpretation that may be worth adding to the paper. The authors imply that the results from this study will be generally applicable (e.g., last paragraph of the Introduction), but I suspect they will be most applicable to systems where genetic architecture of resistance is similar to what is seen in this system: variation in resistance explained by few genes of large effect. I am less confident that systems where variation in resistance is determined by a large set of genes with individually small effects will fit the evolutionary model presented here. The authors interchangeably using the words pathogen and parasite to describe the infection, but viruses are a specialized class. This may collapse to being a model for rapidly evolving specialists with less application to generalists or more slowly evolving pathogens and parasites. The authors themselves emphasize the nearly Mendelian architecture of resistance (paragraph one) and the specialist nature of the coevolutionary interaction (paragraph four) in Discussion and at multiple places in Results. In the final paragraph of the Discussion, they conclude that selection by viruses increases genetic diversity and causes a simple genetic architecture of resistance, but I suspect the causality is the other way around: a simple architecture of resistance and rapid evolution of the pathogen facilitates the evolved increase in diversity.

Upon reading Figure 2, I wondered if there are some host families that are universally susceptible and others that are universally resistant to the non-coevolved viruses. This would be evident in the correlation among families in viral load of the two non-coevolved viruses for each species. Those correlations are given in Supplementary file 3 but they are never discussed in the main text. I would add a few sentences describing these correlations and their implication, perhaps in subsection “Major-effect genetic variants that are known to provide resistance to DMelSV do not protect against other viruses” where the correlation in resistance between coevolved and non-coevolved viruses is presented. This could also be evaluated by looking at the correlation in resistance to DAffSV and DObsSV across the DSPR, which is not presented. In Supplementary file 3, the correlation in resistance to non-coevolved viruses is stronger in *D. immigrans* and *D. subobscura* and is also fairly strong in *D. melanogaster*. This would appear to support variation for generalized resistance in the absence of specific coevolution.

*Reviewer #2:*

The major result of this study is that organisms harbor more genetic variation to deal with their native pathogens than with foreign ones. This is interesting and important because it tells us about the nature of genetic variation for pathogen resistance. It is also informative with respect to predicting how hosts might respond when infected with novel pathogens. The paper is very nicely written, interesting and easy to follow, and the authors have done a very thorough job and used a number of interesting approaches. They start by infecting four species of *Drosophila* with native and foreign Sigma viruses. All four show the same pattern – significantly more genetic variation in the titer of native virus. The rest of the paper focuses on *D. melanogaster*. The authors show that variation in two genes previously shown to confer resistance to native Sigma virus don't have a major effect on foreign Sigma viruses. They also perform an additional QTL analysis of resistance, identifying 7 QTL (only one of these had been previously shown to be imprortant in virus resistance, so these new QTLs will be interesting to follow up on). Most of the QTLs did not affect titers of the foreign viruses.

My main concern is that I am not sure that this study is so different from previous work. The authors performed a similar study, with a similar conclusion, published in PLoS Genetics (Magwire et al., 2012). In that study they assessed genetic variation in *D. melanogaster* resistance to 2 native and 2 foreign viruses (including two viruses used here – DMelSV and DAffSV). It would be useful for the authors to discuss this paper and how it compares with the present one more explicitly. The authors have also done extensive work identifying genetic variants that contribute to resistance to Sigma virus (and other viruses) in *D. melanogaster*.

It would also be useful for the authors to include some more information and details about Sigma virus. How does Sigma virus affect host fitness (the authors don't say much about this except that there was no mortality after 15 days)? It is also a bit hard to tell how successful novel infections are. What percent of individuals/families are not infected at all? (It looks like the infection rate of DAffSV in *D. melanogaster* is quite low in the Magwire et al. paper.) What is known about genetic variation in tolerance to Sigma viruses?

*Reviewer #3:*

This is a very nice paper showing that pathogens select for not just resistance per se but specifically that they can select for increased genetic variation in resistance in host populations. Moreover, they can select for genes/gene-action of major effect. The analysis is carried out using large cross inoculations studies, inoculations on lineages with known resistance genes in *D. melanogaster*, and by a QTL study of a composite inbred population…and the results are congruent among all approaches.

My only (somewhat apologetic) gripe is that one of our papers which showed essentially the same results (selection for genes of large effect, increased variance, and increased pathogen infectivity) based on a reanalysis of a large cross-inoculation experiment in the Linum-Melampsora system, is not cited (Antonovics et al., 2011). However, what we bemoaned in that paper (lack of knowledge of the genes involved) is very much made up for in this paper.

I also think it would be useful to have clear discussion of the previous theoretical work by the Boots group showing that there are conditions under which co-evolution would indeed result in increased variance in resistance, yet that there are equally other conditions under which the opposite result of reduced variance is expected. An application of this theory to levels of genetic variation in susceptibility to different human diseases is in Baker and Antonovics, 2012. Basically, it would be nice to strengthen the Discussion, paragraph two, and get a better sense of whether the authors think that the inference that changing selection pressures, paragraph four, result in increased variance is sound and parsimonious – or if it is just guessing.

The paper is focused on viral loads, but what the effect is of these loads on fitness is never clear, and some mention of this should be included in the discussion if not before.

---

## [Author Response]

[Editors’ note: the author responses to the first round of peer review follow.]

As you will note, reviewer's comments were generally positive about your manuscript. Nevertheless, in the course of the discussion between reviewers, two issues arise that justify the decision:1) The present paper was judged too similar to a previous study of the authors (see reviewer 2). It was felt that the authors have already shown this result with melanogaster and that the present study is expanding on that previous observation using more host species. Thus the novelty was questioned.2) The generality of the result might be overstated. Reviewers felt that the authors over-claim the implication for general host-parasite evolution, and that the pattern they described might only apply to infections for which resistance has a simple genetic architecture and the pathogen is rapidly evolving, although that may capture most specialist viruses.

We understand the reasons why it was rejected, but would ask you to reconsider the manuscript for two reasons.

The first reason for the manuscript being rejected was that the conclusions are similar to one of our earlier papers (Magwire et al., 2012). This was the inspiration for this study, but here we have rigorously tested what were anecdotal conclusions in that study. The number of host-virus associations in the Magwire paper was simply too few to draw robust associations between coevolution and levels of genetic variation. We now have a paragraph in the Introduction discussing the results of Magwire, 2012 and explaining how our new data advances on this. We argue that the Magwire paper dataset generated the hypothesis we are testing, and the ability to build on previous work underpins rigorous and reproducible science.

The second reason for rejection was that our conclusions might not be applicable to other classes of parasites. Rereading the manuscript it was clear our choice of words overstated the generality of our results, and we are now much more careful with our prose. We also discuss the extent to which our conclusions may be extrapolated to other viruses and pathogens by a more detailed review of the literature. All work on model organisms aims to shed light on biology more broadly, but whether it succeeds ultimately awaits research on other species.

Reviewer #1:[…] I have one major point of interpretation that may be worth adding to the paper. The authors imply that the results from this study will be generally applicable (e.g., last paragraph of the Introduction), but I suspect they will be most applicable to systems where genetic architecture of resistance is similar to what is seen in this system: variation in resistance explained by few genes of large effect. I am less confident that systems where variation in resistance is determined by a large set of genes with individually small effects will fit the evolutionary model presented here. The authors interchangeably using the words pathogen and parasite to describe the infection, but viruses are a specialized class. This may collapse to being a model for rapidly evolving specialists with less application to generalists or more slowly evolving pathogens and parasites. The authors themselves emphasize the nearly Mendelian architecture of resistance (paragraph one) and the specialist nature of the coevolutionary interaction (paragraph four) in Discussion and at multiple places in Results. In the final paragraph of the Discussion, they conclude that selection by viruses increases genetic diversity and causes a simple genetic architecture of resistance, but I suspect the causality is the other way around: a simple architecture of resistance and rapid evolution of the pathogen facilitates the evolved increase in diversity.

Reading our manuscript again we agree our choice of words overstated the generality of our conclusions. Throughout the manuscript we have altered our language to make it clear our results relate to a group of *Drosophila* viruses, and have added three new paragraphs to the Discussion relating our results to the literature. These changes include:

1) We have changed the text throughout, including the Abstract, last paragraph of the Introduction, and first paragraph of the Discussion to refer to viruses/sigma viruses and avoid the impression our results necessarily apply to all parasites.

2) We have changed all instances of parasite to pathogen, virus or sigma virus.

3) We have added these paragraphs to the Discussion:

“A key question is whether the increased genetic variation that we see in coevolved *Drosophila*–sigma virus interactions will hold for coevolved pathogens more generally. […] Nonetheless, in an analogous study of wild flax plants, there was less genetic variation in susceptibility to sympatric (coevolved) fungal pathogens [Antonovics et al., 2011].”

“Regardless of its causes, it may be common that susceptibility to infectious disease has a simple genetic basis. […] In these studies the polymorphism with the largest effect was found against the only natural *D. melanogaster* pathogen tested (a polymorphism in Diptericin detected using Providencia rettgeri infection), anecdotally supporting the patterns seen here [Unckless, Rottschaefer and Lazzaro, 2015].”

Upon reading Figure 2, I wondered if there are some host families that are universally susceptible and others that are universally resistant to the non-coevolved viruses. This would be evident in the correlation among families in viral load of the two non-coevolved viruses for each species. Those correlations are given in Supplementary file 3 but they are never discussed in the main text. I would add a few sentences describing these correlations and their implication, perhaps in subsection “Major-effect genetic variants that are known to provide resistance to DMelSV do not protect against other viruses” where the correlation in resistance between coevolved and non-coevolved viruses is presented. This could also be evaluated by looking at the correlation in resistance to DAffSV and DObsSV across the DSPR, which is not presented. In Supplementary file 3, the correlation in resistance to non-coevolved viruses is stronger in *D. immigrans* and *D. subobscura* and is also fairly strong in *D. melanogaster*. This would appear to support variation for generalized resistance in the absence of specific coevolution.

We have added the DSPR genetic correlations, which are fairly well-estimated, to Supplementary file 3.

Testing whether resistance to the non-coevolved viruses tends to be ‘non-specific’ is good suggestion which our experimental design and analysis was designed to address. Unfortunately, it was frequently not possible to inject the same family with multiple viruses, so the correlations mostly have broad confidence intervals (Supplementary file 3). Especially now the DSPR results are added; this means we cannot reach any robust conclusion as to whether genes that confer resistance to the non-coevolved viruses tend to be less specific in their action. For this reason we restricted our results to highlighting the one very clear result – different genes control resistance to different viruses. We now draw the reader’s attention to the non-coevolved pairs:

“(Supplementary file 3; note correlations are frequently low between pairs of non-endemic viruses too).”

Reviewer #2:The major result of this study is that organisms harbor more genetic variation to deal with their native pathogens than with foreign ones. This is interesting and important because it tells us about the nature of genetic variation for pathogen resistance. It is also informative with respect to predicting how hosts might respond when infected with novel pathogens. The paper is very nicely written, interesting and easy to follow, and the authors have done a very thorough job and used a number of interesting approaches. They start by infecting four species of Drosophila with native and foreign Sigma viruses. All four show the same pattern – significantly more genetic variation in the titer of native virus. The rest of the paper focuses on *D. melanogaster*. The authors show that variation in two genes previously shown to confer resistance to native Sigma virus don't have a major effect on foreign Sigma viruses. They also perform an additional QTL analysis of resistance, identifying 7 QTL (only one of these had been previously shown to be imprortant in virus resistance, so these new QTLs will be interesting to follow up on). Most of the QTLs did not affect titers of the foreign viruses.My main concern is that I am not sure that this study is so different from previous work. The authors performed a similar study, with a similar conclusion, published in PLoS Genetics (Magwire et al., 2012). In that study they assessed genetic variation in *D. melanogaster* resistance to 2 native and 2 foreign viruses (including two viruses used here – DMelSV and DAffSV). It would be useful for the authors to discuss this paper and how it compares with the present one more explicitly. The authors have also done extensive work identifying genetic variants that contribute to resistance to Sigma virus (and other viruses) in *D. melanogaster*.

Our response to this criticism is in our covering letter. We have added a paragraph to the Introduction that explains how this dataset advances Magwire, 2012:

“As part of a whole genome association study, we have previously estimated levels of genetic variation in the susceptibility of *D. melanogaster* to four different viruses [Magwire et al., 2012]. […]For this reason, here we return to this question and formally test whether a history of coevolution alters the amount and nature of genetic variation.”

It would also be useful for the authors to include some more information and details about Sigma virus. How does Sigma virus affect host fitness (the authors don't say much about this except that there was no mortality after 15 days)?

The two most reliable and relevant studies of the effect of the virus on the host have estimated its effect on host fitness without estimating fitness components. Despite using entirely different methods these generated very similar results. Nonetheless, it seems it is unlikely the cost of infection is due to adult mortality. The Introduction now reads:

“In *Drosophila melanogaster*, despite the virus causing little adult mortality, infection reduces host fitness by approximately 25% [Fleuriet, 1988; Wilfert and Jiggins, 2010].”

It is also a bit hard to tell how successful novel infections are. What percent of individuals/families are not infected at all? (It looks like the infection rate of DAffSV in *D. melanogaster* is quite low in the Magwire et al. paper.)

We could detect viral RNA in all samples, and our previous data (Longdon et al 2011 PLOS Pathogens) suggest the virus must be replicating at some level to persist in the host for 15 days. Note Magwire et al. measured a symptom of sigma virus infection (CO2 sensitivity) that represents whether the infection is over a threshold viral load. The far greater dynamic range of the quantitative PCR assay we used is a considerable improvement over Magwire et al.’s approach.

What is known about genetic variation in tolerance to Sigma viruses?

Nothing is known. We have never measured tolerance, and the resistance genes studied in *D. melanogaster* all act to reduce viral load (see Figure 3 and Cao et al., 2015)

Reviewer #3:This is a very nice paper showing that pathogens select for not just resistance per se but specifically that they can select for increased genetic variation in resistance in host populations. Moreover, they can select for genes/gene-action of major effect. The analysis is carried out using large cross inoculations studies, inoculations on lineages with known resistance genes in *D. melanogaster*, and by a QTL study of a composite inbred population…and the results are congruent among all approaches.My only (somewhat apologetic) gripe is that one of our papers which showed essentially the same results (selection for genes of large effect, increased variance, and increased pathogen infectivity) based on a reanalysis of a large cross-inoculation experiment in the Linum-Melampsora system, is not cited (Antonovics et al., 2011). However, what we bemoaned in that paper (lack of knowledge of the genes involved) is very much made up for in this paper.

Sorry we missed this; we now discuss this at two places in the Discussion (paragraphs three - five).

I also think it would be useful to have clear discussion of the previous theoretical work by the Boots group showing that there are conditions under which co-evolution would indeed result in increased variance in resistance, yet that there are equally other conditions under which the opposite result of reduced variance is expected. An application of this theory to levels of genetic variation in susceptibility to different human diseases is in Baker and Antonovics, 2012. Basically, it would be nice to strengthen the Discussion, paragraph two, and get a better sense of whether the authors think that the inference that changing selection pressures, paragraph four, result in increased variance is sound and parsimonious – or if it is just guessing.

The sigma virus system is unusual for an animal pathogen in that we have a strong literature on its genetics and how selection has acted. We have therefore tried to keep our discussion strongly rooted in this. In particular there is a recurrent pattern if selective sweeps, and we have highlighted population genetics theory about how this can affect genetic variation (note that the adaptive dynamics models of Boots are not appropriate to model these processes). Alongside this we have now considerably increased our discussion of the coevolution literature, including the Boots models. The changes are in the two paragraphs quoted at the start of our response to Reviewer 1.

The paper is focused on viral loads, but what the effect is of these loads on fitness is never clear, and some mention of this should be included in the discussion if not before.

Unfortunately, it was not feasible to measure the fitness of >50,000 flies. Fortunately, there is good reason to believe viral load correlated with fitness.

1) The virus reduces fitness, and viral load is correlated with vertical transmission rates and infection rates in both in the lab and field

2) Alleles of two genes that reduce viral loads have been positively selected in nature.

We have added the text:

“Studies on DMelSV have shown that loci that reduce loads when the virus is injected also reduce infection rates in both the lab and field [2, 27, 43-46]. As infection is costly, this is expected to increase host fitness.”